# Prevalence and prediction of Lyme disease in Hainan province

Lin Zhang[1], Xiong Zhu[2], Xuexia Hou[1], Huan Li[2], Xiaona Yang[1], Ting Chen[2], Xiaoying Fu[2], Guangqing Miao[3], Qin Hao[1]*, Sha Li[2]*

**1** State Key Laboratory for Infectious Disease Prevention and Control, National Institute for Communicable Disease Control and Prevention, Chinese Center for Disease Control and Prevention, Beijing, China, **2** People's Hospital of Sanya, Hainan province, China, **3** Affiliated Hospital of Hangzhou Normal University, Hangzhou, China

☯ These authors contributed equally to this work.
* haoqin@icdc.cn (QH); lisha8502@126.com (SL)

**Data Availability Statement:** All relevant data are within the manuscript and its Supporting Information files.

**Funding:** This work was supported by The Key Research & Development Program of Hainan

## Abstract

Lyme disease (LD) is one of the most important vector-borne diseases worldwide. However, there is limited information on the prevalence and risk analysis using correlated factors in the tropical areas. A total of 1583 serum samples, collected from five hospitals of Hainan Province, were tested by immunofluorescence assay (IFA) and western blot (WB) analyses using anti-*Borrelia burgdorferi* antibodies. Then, we mapped the distribution of positive rate (by IFA) and the spread of confirmed Lyme patients (by WB). Using ArcGIS, we compiled host-vector-human interactions and correlated data as risk factor layers to predict LD risk in Hainan Province. There are three LD hotspots, designated hotspot I, which is located in central Hainan, hotspot II, which contains Sanya district, and hotspot III, which lies in the Haikou-Qiongshan area. The positive rate (16.67% by IFA) of LD in Qiongzhong, located in hotspot I, was higher than that in four other areas. Of confirmed cases of LD, 80.77% of patients (42/52) whose results had been confirmed by WB were in hotspots I and III. Hotspot II, with unknowed prevalence of LD, need to be paid more attention considering human-vector interaction. Wuzhi and Limu mountains might be the most important areas for the prevalence of LD, as the severe host-vector and human-vector interactions lead to a potential origin site for LD. Qiongzhong is the riskiest area and is located to the east of Wuzhi Mountain. In the Sanya and Haikou-Qiongshan area, intervening in the human-vector interaction would help control the prevalence of LD.

## Author summary

Lyme disease (LD) is an enzootic disease that is widely distributed in Asia, America, and Europe. In recent years, the incidence and the endemic range of LD have steadily expanded with increased human influence. Hainan Province has a tropical climate and is located in the South China Sea, and the first cases of LD in Hainan province were reported in 2015. However, there is lack of information on the prevalence and risk distribution of LD on the island. To analyze the prevalence of LD, we examined 1583 serum samples

Province (ZDYF2017163), to XZ; The National Critical Project for Science and Technology on Infectious Disease of P.R.China(2016ZX10004001-004 and 2017ZX10303404-006-003), to QH; Medicine & Health Program of Hainan Province (1602320116A2001), to XZ. The funders had no role in study design, data collection and analysis, decision to publish, or preparation of the manuscript.

**Competing interests:** The authors have declared that no competing interests exist.

from patients presenting with arthritis and neurological symptoms. We then mapped the risk of LD using correlated factors such as the distribution of Muridae and Ixodidae, and human activity. This study is the first to demonstrate the distribution of LD risk in Hainan Province. A better understanding of the host-vector and human-vector interactions, as well as distribution of high-risk areas will serve as a good starting point to prevent this disease.

## Introduction

Lyme disease (LD) is caused by the tick-borne spirochete *Borrelia burgdorferi*, and has been reported to be widely distributed across China mainland[1–2]. Human LD generally occurs in stages, from the early localized stage of erythema migrans, fatigue, chills, and fever, to a late disseminated stage of intermittent bouts of arthritis with severe joint pain and swelling and neurological symptoms [3–4].

While culture of *B. burgdorferi* from patient specimens is difficult and single serological tests are partially hampered by the disease stage at which antibodies start to appear, leading to insufficient results; thus, patients may still be seronegative in the early stages of infection[4–5]. In China, without surveillance, Lyme patients often don't find themselves ill until late symptoms occur. Therefore, we only collected serum samples from patients with arthritis and neurological symptoms, as these two symptoms were common in lyme patients[6]. Nowadays, the US-CDC diagnosed LD by serological tests in a two-step process, including a screening test (enzyme immunoassay (EIA) or immunofluorescence assay (IFA)) and confirmation test (western blotting, WB)[4].

LD is known to be a vector-borne disease distributed by ticks[7]. The spirochete preserved in nature depends on transmission through the host-vector route[8–9]. Tick bites are the most common way by which humans acquire LD[10]. The abundance of tick communities has deep implications for LD[11]. The risk of acquiring this disease depends on the encounter of ticks that are bacterial carriers with human beings at the appropriate site[12]. This process involves several complicated factors, including the presence of a suitable habitat for tick survival, breeding, and questing, and a proper area where humans can come into contact with questing ticks. While the relationship between the incidence and density of infected nymphs is sometimes weak [13,14], it is due to the complicated ecosystem for their unknown interactions. This gap could be partly filled by predictive models. Currently, there are no vaccines that can protect humans from LD; therefore, the appropriate action is necessary to control the risk of infection to avoid tick exposure [15,16]. To effectively target prevention and control, public organizations need spatial estimates of LD risk, and correlated factors should be compiled in order to model disease prevalence.

LD infection mostly occurs after a tick bite. In rural areas, locals are bitten by ticks in farmland, grassland, bush, and forest in their daily life[17]. The risk of acquiring LD depends on ticks that may or may not be carrying the pathogen *B. burgdorferi*. Hence, it is clear that ticks probably carry the *borrelia burgdorferi* through contact with their hosts[18]. A previous study revealed that small mammals are the most important hosts of ticks, and play an essential role in persistence of *B. burgdorferi* [19]. In China, *B. burgdorferi* strains have been isolated either from ticks or from the urinary bladders or kidneys of rodents of the family Muridae [20,21].

Hainan (also Hainan Island) is located in the South China Sea. It is separated from Guangdong's Leizhou Peninsula to the North by Qiongzhou Strait. The island is renowned for its tropical climate, which is completely different from that of the Chinese mainland. Due to its

unique geographical situation and climate, it is important to study the distribution of LD in Hainan Province, as the epidemiology, pathogenesis, and management of LD in Hainan would be different from those in the Chinese mainland. Tick borne disease spotted fever was firstly discovered since 1990s in Hainan[22]. However, there is still lack of data about LD in Hainan Province. Then, we investigated and confirmed that LD existed in Hainan Province since 2014 [23]. All six patients were distributed in the south coast of Hainan Province. Furthermore, the investigation was insufficient for the lack of research in the central and northern regions of Hainan Province. Therefore, the serums in our investigation were from 5 hospitals which were located in the central and north Hainan (Fig 1). Temperature regulation of the development rates and mortality of ticks depends on water loss, as these factors are regulated by the relative humidity and the air saturation deficit[24]. Therefore, from September to November, an increasing number of ticks appear outside and the natural environment during this time in Hainan is favorable to ticks. Normally, along with growth in number of ticks, patients with LD show a similar increasing a same trend with a time lag[11]. To study the distribution of patients with LD and the potential areas that are at risk for acquiring LD, and to understand the spread of LD in Hainan Province, we conduct investigations in suspected patients presenting with LD-related symptoms. Here, from September to November, we collected blood samples from patients in five hospitals presenting with arthritis and nervous system disease which were all the large general hospitals in local area. Then, we analyzed the spatial pattern of LD risk using the knowledge based method with the correlated factors.

## Methods

### Ethics statement

This study and the research protocol were reviewed and approved by the Ethical Committee (Institutional Review Board, IRB) of People's Hospital of Sanya (License number: ME[2018]-3). All patients gave written informed consent for participation in this study with their identifiable information, and the legal guardians of young children (less than 12 years of age) provided informed consent on their behalf; in accordance with the Declaration of IRB approval.

### Blood samples

We collected 1583 serums from patients with arthritis or neurological symptoms from 5 hospitals in Hainan province of China from September to November in 2015. Five hospitals were Wenchang People's Hospital, Dan zhou People's Hospital, Dongfang People's Hospital, Qiongzhong People's Hospital and The Third People's Hospital of Haikou (Fig 1). These samples were examined for the presence of *B. burgdorferi* serologically by the two-step tests (IFA and WB).

### Immunofluorescent assay (IFA)

IFA method was used to screen serum samples for anti-Bb (*Borrelia burgdorferi*) antibodies by antigen slides as a surrogate antigen[1]. Serums were initially screened at dilutions of 1:128 for IgG antibody and 1:64 for IgM antibody. Titers of positive serum samples were subsequently determined to end point.

Antigen used to test anti-*B. burgdorferi s. l.* IgG antibodies was prepared from a Chinese human *B. garinii* isolate, PD91. The PD91 isolate was cultivated in BSK-media at 33°C for a week, harvested and washed in phosphate-buffered saline. Antigen was spotted onto the wells of microtiter slides and fixed with acetone. A titer of ≥1/128 for IgG or 1:64 for IgM was considered positive [23].

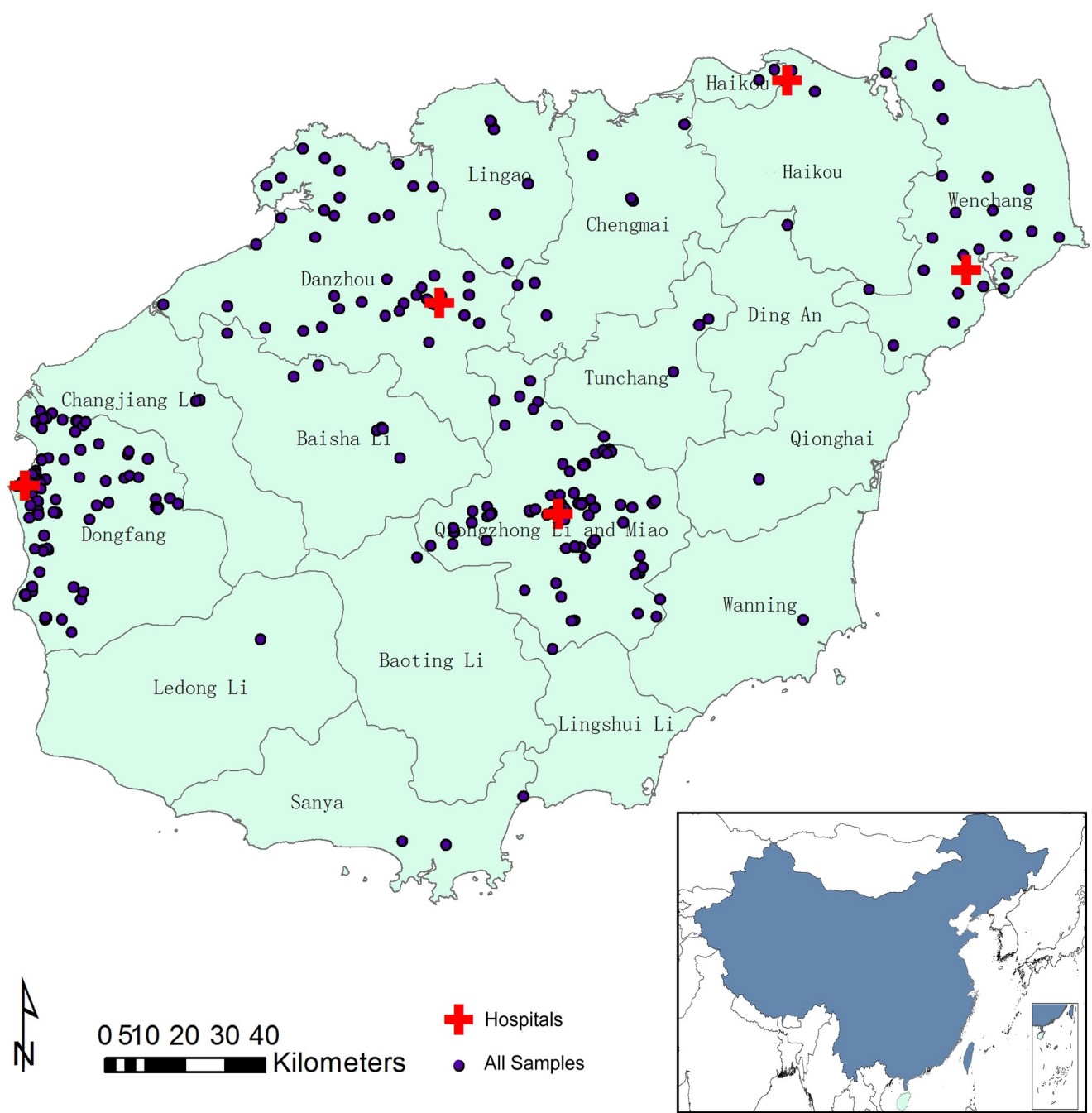

**Fig 1. Distribution of 1583 serum samples from five hospitals.** This map was plotted by a combination of ArcGIS software version 10.1 (Esri, USA) and Photoshop CS 2.0 software (Adobe Systems, USA).

### Western Blot (WB)

With the positive or equivocal results of IFA test yielded, we conducted both the IgG and IgM Western Blot (WB) assay for further confirmation.

Antigen strips (provided by China CDC, use *B.garinii* PD91 as an antigen) were put into the seropositive samples by IFA, which diluted with PBS-T by 1:25 for both IgG and IgM, then incubated at room temperature in Orbital Shaker for 4 hours. After five washes of at least 10

min each, the strips then were incubated over night with horseradish peroxidase-conjugated rabbit anti-human IgG (1:1500) and IgM (1:2500) antibodies (Sigma), respectively. After washing for 10 min with PBST, we used TIANGEN Enhanced HRP-DAB Chromogenic Substrate Kit to develop the color to identify the bands. The color development was stopped by purred water when the positive control serum sample reached a defined intensity.

Three pathogenic genotypes which are *B.garinii*, *B. afzelii* and *Borrelia burgdorferi* sensu stricto exist in China. According to their different genomic species, the United States[25–26] and Europe[27] separately established criteria for a positive WB result. In China, criteria for a positive diagnosis of *B.garinii* were established in 2010 as *B.garinii* was the predominat genotype[21,28]. It was at least one band of P83/100, P58, P39, OspB, OspA, P30, P28, OspC, P17, P66 and P14 in the IgG test and at least one band of P83/100, P58, P39, OspA, P30, P28, OspC, P17 and P41 in the IgM test[29–30].

## Spatial data

We compiled three types of spatial data for each of the 19 counties of Hainan province. These were host data, vector data and socioeconomic ecological data. Here, all data were used in subsequent analyses.

**Host data.**  The wild species list (Muridae) is based on widely used database GBIF(Global Biodiversity Information Facility, https://www.gbif.org/) (Muridae Information see S1 Data). We chose one Familie of rodentia in this study, which is Muridae as the rodents of the family Muridae were the most important reservoir of *B. Burgdorferi* in China[31–34]. We removed a specimen data *Chiropodomys gliroides*, which were historically distributed in Hainan but not reported in nowadays. All the rest records have explicit geographic coordinates. We carefully checked geographic and taxonomic accuracy for each species and excluded those species without geographic information.

**Vector data.**  Data on vector (Ixodidae) distribution were derived from a previous study (Ixodidae Information see S2 Data)[35]. We chose Ixodidae as vector in our prediction because most ixodes need more than one host to finish their life cycle. This habitual nature would facilitate the transmission of disease between hosts[36]. We carefully checked geographic accuracy for each species and deleted repetition data. We mapped the distribution of Ixodidae according to the longitude and latitude. We also calculated the Distance Band and mapped the hotspots of Ixodidae in Hainan Province. The methods are same as Muridae hot spot prediction.

**Socioeconomic ecological data.**  Based on the 2019 Hainan province Statistical Yearbook, we collected county-level data on area of cultivated land, number of cattle in stock, number of goat in stock, area of forest, area of grassland, total value of farming, total value of husbandry, and total value of forestry, area of forest, rural population density. Data on land-cover types were retrieved from Resource and Environment Science and Data Center (http://www.resdc.cn/data.aspx?DATAID=335). The land cover data were derived from the 2020 Landsat8 Land Cover product, Land Cover Type Yearly National 1km, which classifies land cover as 6 primary categories and 25 secondary categories (S1 Table). Land cover types were combined and reclassified into 10 land-use categories, which were rice paddy field, dry land, grassland, bush, forest land, wet land, rural area, urban and construction land, unused land and sea (S1 Table).

We generated *LUmix* layer to demonstrate the area of Muridae and Ixodidae contact. We reclassified the land-use layer which the possible contact area: bush as "9", grassland as "8", dry land as "7", rice paddy field as "6"; the rare contact area: rural area as "5", forest land as "4", urban and construction land as "3", unused land as "2", while we assigned grid value of impossible contact area-lake and sea -as "1". In order to display where the farming activity happens

on map, we generated raster of farmland (LUfa) using reclassify tool in ArcGIS. As 68% rice paddy fields were three ripe cycle land (wet-wet-dry) in Hainan, also based on 8-hour working/about 40 minutes time spent on the way, we assigned the value of the dryland and rural area grids to "12", whereas rice paddy field grids were "4" and other land-use type grids were "1". We also created raster of grassland and bush (*LUh*) layer, where grassland, bush and rural area grids were reclassified as "12" and other grids were "1". The raster of forests(*LUfo*) layer was the reclassified land-use layer where forest land, bush and rural area grids value were assigned as "12" and other grids were "1".(Land-use Reclassification Remap Table see S2 Table)

## Analyses

According to the addresses of suspected cases, we found the latitude and longitude of all 1583 patients we examined. Then we added these data as points into ArcGIS using Display XY Data tool (Fig 1). All layers were using the geographic coordinate system GCS_WGS_1984. It should be noticed that we might need to project all layers using the identical projected system to avoid the distance errors when facing the bigger extent of study area, like country or even continent. As long as the projected system were equal for all layers in risk analysis, using which one doesn't change location of highly risk areas, but only the visual sizes of them displayed on the map. To visualize the positive rate of Lyme disease in 5 study areas, we typed IFA positive rate into the attribute table as a new field of the county-level vector layer of Hainan (Fig 2A). Then we added WB positives as points into the map using the tool Display XY Data(Fig 2B).

Two primary risk assessment analysis were conducted. The first was about Muridae-Ixodidae interactions, incorporated factors relating to the muridae hotspot, and the Ixodidae

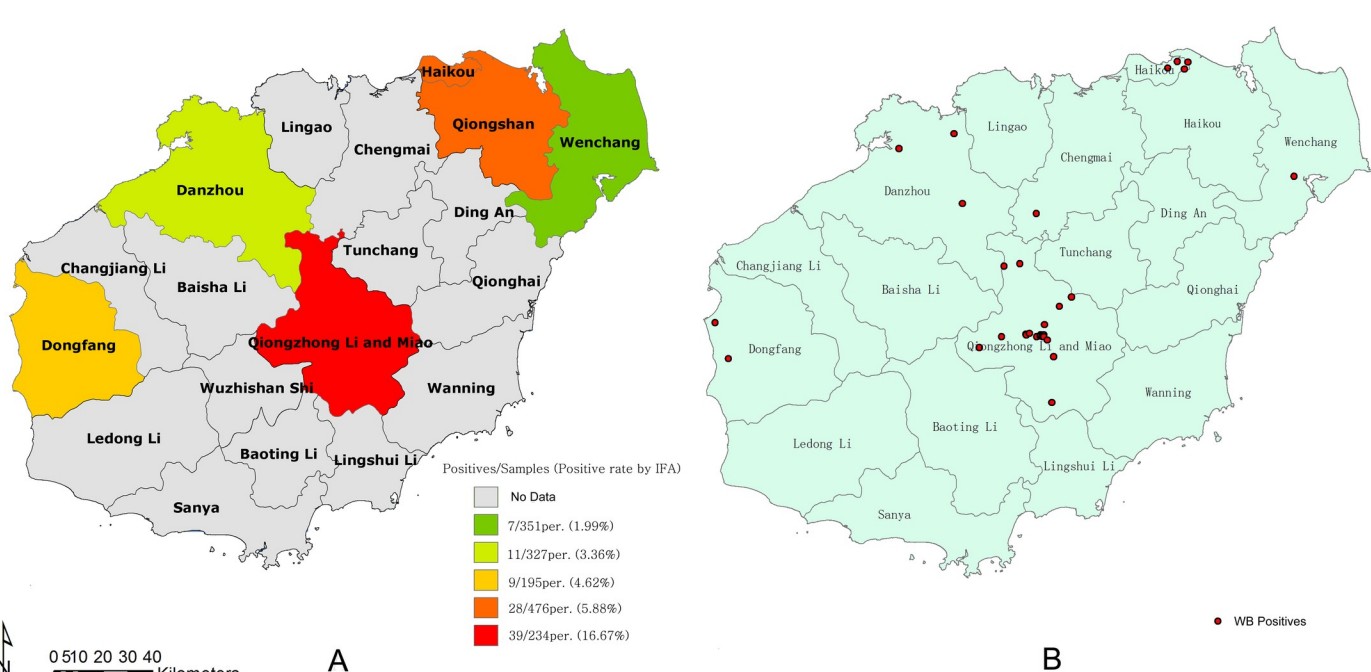

**Fig 2. Distribution of IFA- and WB-positive cases in Hainan Province.** (A) Distribution of the IFA-positive rate in our research in Hainan Province. (B) There were a total of 52 confirmed patients among the 94 IFA-positive cases in Hainan.This map was plotted using a combination of ArcGIS software version 10.1 (ESRI, USA) and Photoshop CS 2.0 software (Adobe Systems, USA).

hotspot. We then generated a risk map incorporated the anthropogenic factors by combining the layers related to Lyme risk that rural people might face tick. Finally, we calculated the sum of these two risk to evaluate the lyme risk in Hainan province.(Models information see S1–S3 Figs)

To eliminate the interference of dimensional differences on the assessment, the area of farmland (*Afa*), the intensity of farming activity (*Ifa*), the raster of farmland (*LUfa*), the area of grassland (*Ah*), the intensity of husbandry (*Ih*), the density of cows (*Dcw*), the density of goats (*Dgt*), the raster of grassland and bush (*LUh*), the intensity of forestry (*Ifo*), the raster of forests (*LUfo*), and the densities of rural population (*Dpr*) were normalized using Fuzzy Membership tool in ArcGIS, respectively. While the area of forests (*Afo*) cannot be normalized using Fuzzy Membership as the counties with no forests were exported to no data. Therefore, we used their own value divided by the maximum value.

## Host hot spot analysis

As research investigation data points (presence data) could not thoroughly reflect the animals' distribution, we generated the hot spot map of Muridae and Ixodidae to demonstrate the clusters of their distribution considering spatial autocorrelation. We created a Grid feature using Create Fishnet tool. The grid size of this feature was $10\times10km$, as the range of rodent's activity were relatively wide driven by food searching (as host movements contribute importantly to tick distribution, especially in lyme disease [37–39], we therefore set the same grid size $10\times10km$ as Muridae in Fishnet Feature). Then we calculate the Muridae and Ixodidae distribution in every fishnet cell using Spatial Join tool, respectively. At the same time, we calculated the Distance Band using Multi-Distance Spatial Cluster Analysis tool (Ripleys K) to the point layer of distribution of Muridae and Ixodidae, respectively. Based on the Spatial Join of Muridae and Ixodidae to fishnet layers, we mapped the hotspots of Muridae ($Gm_i^*$) and Ixodidae ($Gix_i^*$) by using the Hot Spot Analysis tool to calculate the Getis-Ord Gi* statistic in ArcGIS respectively (details see ArcGIS Resource Center. http://resources.esri.com/help/9.3/ ArcGISEngine/java/Gp_ToolRef/Spatial_Statistics_tools/how_hot_spot_analysis_colon_ getis_ord_gi_star_spatial_statistics_works.htm). The Getis-Ord General G tool is an inferential statistic and is most appropriate looking for unexpected spatial spikes of high values. We used the largest DiffK value (Distance Band exported, S3 Table) as the appropriate Fixed Distance when running the Hot Spot Analysis, as the largest DiffK value reflecting the distance where spatial processes promoting clustering are most pronounced.

$$G_i* = \frac{\sum_{j=1}^{n} w_{i,j}x_j - \bar{X}\sum_{j=1}^{n} w_{i,j}}{s\sqrt{\frac{\left[n\sum_{j=1}^{n} w_{i,j}^2 - \left(\sum_{j=1}^{n} w_{i,j}\right)^2\right]}{n-1}}},$$

where $x_j$ is the total number of Muridae (Ixodidae in tick Hot Spots Analysis) distribution for each fishnet feature cell *j*, $w_{i,j}$ is the spatial weight between feature *i* (the feature class for which Hot Spots Analysis will be performed) and *j* (Muridae distribution feature, Ixodidae distribution feature when calculating tick hotspots), n is equal to the total number of features and:

$$\bar{X} = \frac{\sum_{j=1}^{n} x_j}{n},$$

$$s = \sqrt{\frac{\sum_{j=1}^{n} x_j^2}{n} - (\bar{X})^2},$$

### Lyme risk from host-tick interactions

Contacts between hosts and vectors are principal pathways of Borrelia burgdorferi transmission [7,8], as well as most of tick-borne diseases. We assessed Lyme risk from potential contact between Muridae and Ixodidae by mapping the distribution hotspots of these two genera.

Contact rate between Muridae and Ixodidae are difficult to quantify and there have been few attempts to do so. While we could assess the risk by overlapping the distribution hotspots of Muridae and Ixodidae, which assuming that interlap of these two genera possible distribution revealed the most risk of contacts. In the grass land and forest area, small mammals and ticks probably come into contact through tick life history. The land-use layer was used to indicate the spatial patterns of these potential contact zones. Therefore, we overlapped the hotspots of Muridae ($Gm_i^*$), the hotspots of Ixodidae ($Gix_i^*$) and reclassified land-use layer ($LUmix$) using Fuzzy Overlay tool in ArcGIS.

To assess the Lyme risk from the contact between Muridae and Ixodidae, we generated the Muridae-Ixodidae layer, in which the risk value of $i$th cell, $Rmix_i$, was calculated as:

$$Rmix_i = Gm_i^* Gix_i^* LUmix_i,$$

where $Gm_i^*$ is the Getis-Ord Gi* statistic for Muridae Hot Spot within the $i$th cell and $Gix_i^*$ is for Ixodidae Hot Spot, respectively; and $LUmix_i$ is the potential contact area of Muridae and Ixodidae within the $i$th cell.

### Lyme risk from human-tick interaction

Lyme disease caused by the tick bite is usually happened in bush, grass land or forest area[37]. People who lived in rural area might encounter ticks in their daily life. Due to cultivation, grazing and forestry activity, close contact between ticks and humans can occur on farms, grass land, or forest respectively [40–42]. We carefully classified the kind of human-tick contact in rural, according to people's career and their workplace. These were farming activity, husbandry and forestry. Corresponding to these activities, we used primary variables to generate three secondary variables: (i) coefficient of farming, (ii) coefficient of husbandry, (iii) coefficient of forestry.

With farming activity, farmland served as human-tick contact place. We used area of farmland to quantify the chance of contact, which assuming that the larger area might lead to the bigger chance of contact. The intensity of farming activity in one place revealed the ratio of the value of farming to the sum value of farming, husbandry and forestry. In order to demonstrate the area where human-tick contact happened when farming activity occur, we combined the *Lufa* layer in ArcGIS. To eliminate the interferences of dimensional differences on the assessment, the area of farmland, the intensity of farming activity and *LUfa* layer were normalized using fuzzy Membership Tool in ArcGIS. Then we generate the coefficient of farming activity ($Cfa_i$) by combining these three layers using the fuzzy overlay tool in ArcGIS.

To assess the risk from farming activity, we generated the coefficient of farming activity layer, in which the value of $i$th cell, $Cfa_i$, was calculated as:

$$Cfa_i = Afa_i Ifa_i LUfa_i,$$

$$Ifai_i = \frac{Vfa_i}{Vs_i},$$

where $Afa_i$ is the area of farmland within the $i$th cell and $Ifa_i$ is the intensity of farming activity within the $i$th cell, which is the ratio of the total value of farming to the sum value of farming, husbandry and forestry; $LUfa_i$ is the land-use layer to map where is farming activity happened.

With husbandry, we also calculated the area of grassland and bush [43], and the intensity of husbandry, and reclassified the land use. Moreover, considering the cows and the goats served as the host of ticks [44], we added the density of cows and goats to the equation. We also deal with all 5 layers using Fuzzy Membership, and then we overlapped these layers to calculate the coefficient of husbandry activity ($Ch_i$).

To assess the risk from husbandry, we generated the coefficient of husbandry layer, in which the value of $i$th cell, $Ch_i$ was calculated as:

$$Ch_i = Ah_i Ih_i Dcw_i Dgt_i LUh_i,$$

$$Ih_i = \frac{Vh_i}{Vs_i},$$

where $Ah_i$ is the area of grassland within the $i$th cell and $Ih_i$ is the intensity of husbandry within the $i$th cell, which is the ratio of the total value of husbandry to the sum value of farming, husbandry and forestry; $Dcw_i$ and $Dgt_i$ are the density of cows and goats in stock within the $i$th cell respectively; $LUh_i$ is the land-use layer to map where the grassland and bush are.

With forestry industry, we also calculated the area of forest and the intensity of forestry, and reclassified the land use. We also used Fuzzy Membership to normalize all these layers, and then we overlapped these layers to calculate the coefficient of forestry activity ($Cfo_i$).

To assess the risk from forestry, we generated the coefficient of forestry layer, in which the value of $i$th cell, $Cfo_i$ was calculated as:

$$Cfo_i = Afo_i Ifo_i LUfo_i,$$

$$Ifo_i = \frac{Vfo_i}{Vs_i},$$

where $Afo_i$ is the area of forest within the $i$th cell, containing broadleaved evergreen forest, needle leaved evergreen forest, and $Ifo_i$ is the intensity of forestry within the $i$th cell, which is the ratio of the total value of forestry to the sum value of farming, husbandry and forestry; $LUfo_i$ is the land-use layer to map where the forest are.

Human population density in rural area and distribution of Ixodidae were used to estimate the intensity of these contacts, assuming that higher densities probably result in higher intensities of contacts. In rural area, people will have a greater chance of getting contact with ticks when field operations like farming, husbandry or forestry activities go through.

To assess the Lyme risk from the contact between human and Ixodidae, we generated the human-tick layer by Fuzzy overlaying rural population density layer, sum of $Cfa$, $Ch$ and $Cfo$ layer and hotspots of ixodidae layer, in which the risk value of $i$th cell, $Rpr_i$, was calculated as:

$$Rpr_i = Dpr_i(Cfa_i + Ch_i + Cfo_i)Gix_i^*$$

where $Dpr_i$ is human population density in rural area within the $i$th cell and $Cfa_i$, $Ch_i$ and $Cfo_i$ is the coefficient of farming activity, the coefficient of husbandry and the coefficient of forestry

activity respectively within the $i$th cell, and $Gix_i^*$ is the Getis-Ord Gi* statistic for Ixodidae Hot Spot within the $i$th cell.

## Lyme risk from human-nature interactions

Host communities are thought to be one of the most essential factors on the tick life cycle[45]. They have deep implications for the circulation of tick-transmitted pathogens, because the relative abundance of potential reservoir hosts may produce large variations in the prevalence of such pathogens[46]. For persistence of lyme borreliosis in the field, a seasonal synchronicity among the questing stages of vectors and the abundance of reservoir hosts is also necessary [47]. The risk to humans of lyme disease is proportional to the risk of contact, which not only depends on human-tick contact, but also on the host-tick contact. Therefore, we demonstrate the lyme risk from human-nature interactions, $RR_i$, was calculated as:

$$RR_i = Rmix_i + Rpr_i,$$

where $Rpr_i$ is risk from small mammal-tick interactions within the $i$th cell and $Cfa_i$, $Ch_i$ and $Cfo_i$ is risk from human-tick interactions within the $i$th cell.

## Results

### Origin of suspected patients

A total of 1583 blood samples were collected from five hospitals and added to a map using Arc-GIS according to patients' addresses. Our samples were distributed mostly in the north and central all across Hainan Island (Fig 1).

### IFA test results

A total of 1583 serum samples obtained from patients with arthritis or neurological symptoms were examined for *B. burgdorferi* using the two-step test. A total of 94 serum samples were found to be positive by IFA (IFA titer, 128 for IgG and 64 for IgM); therefore, an overall infection rate of all five hospitals in Hainan Province was 5.94%. The median age of the 94 patients was 53.17 years (range, 13–86 years) and 57.45% of the patients were male. In all the five hospitals, Qiongzhong People's Hospital, which is located in the center of Hainan Province, was the area with the greatest prevalence of LD in our investigation, with 39 IFA-positive cases (positive rate, 16.67%). Then, the Third People's Hospital of Haikou had the second highest prevalence, with a positive rate of 5.88%, followed by Dongfang People's Hospital, Danzhou People's Hospital, and Wenchang People's Hospital with positive rates of 4.62%, 3.36%, and 1.99%, respectively (Table 1).

**Table 1. IFA results of the tested serum samples.**

| Hospital | Serum sample | IFA-positive | Positive rate (%) |
|---|---|---|---|
| Wenchang People's Hospital | 351 | 7 | 1.99 |
| Danzhou People's Hospital | 327 | 11 | 3.36 |
| Dongfang People's Hospital | 195 | 9 | 4.62 |
| Qiongzhong People's Hospital | 234 | 39 | 16.67 |
| Third People's Hospital of Haikou | 476 | 28 | 5.88 |
| Total | 1583 | 94 | 5.94 |

**Table 2. Results of the IFA-positive serum samples tested by WB.**

| Hospitals | Serum samples | IFA-positive | WB-positive | Two-step positive rate (%) |
|---|---|---|---|---|
| Wenchang People's Hospital | 351 | 7 | 4 | 1.14 |
| Danzhou People's Hospital | 327 | 11 | 6 | 1.83 |
| Dongfang People's Hospital | 195 | 9 | 2 | 1.03 |
| Qiongzhong People's Hospital | 234 | 39 | 28 | 11.97 |
| Third People's Hospital of Haikou | 476 | 28 | 12 | 2.52 |
| Total | 1583 | 94 | 94 | 3.28 |

## WB test results

All the 94 IFA-positive serum samples were tested by a standardized Western immunoblot, which is known to have high specificity for LD. Of the 94 IFA-positive samples, 52 were identified positive by WB; thus, the average positive rate was 3.28%. The median age of the 52 patients was 53.88 years (range, 17–85 years), and 57.69% of the patients were male. Among all the five hospitals, Qiongzhong People's Hospital had the highest positive rate of 11.97% according to the two-step testing approach used in our research (28/234) (Table 2).

## LD risk analysis

The calculation of hot spot of Muridae or Ixodidae was based on z-value and p-value in Getis-Ord statistics. A positive and significant z-value indicates spatial clustering of high density of Muridae and Ixodidae (S3 and S4 Datas). On the interaction assessment map of Muridae-Ixodidae (*Rmix*) in Hainan Province, areas with the highest risk (red cells) were relatively concentrated in the central areas of Hainan, which contained the areas north of Qiongzhong Li and Miao, south of Danzhou, east of Baisha Li, and west of Tunchang(Fig 3A).

The human-Ixodidae interaction assessment map (*Rpr*) revealed that in the influence of anthropogenic factors, the highest risk of LD may occur in the intermediate zone of Hainan, and also in the northeast corner of Hainan, which contained provincial capital- Haikou and Qiongshan District. (Fig 3B).

The combination of *Rmix* and *Rpr*, designated *RR*, revealed the risk of humans acquiring LD from human-nature interactions, revealed two hotspots (hotspot I, II and III)(Fig 3C). Hotspot I was located in the central region of Hainan Province, close to the Limu and Wuzhi mountains where the landcover is primarily composed of bush and forests. Hotspot I covered five counties. Hotspot II was in the south Hainan, which contained Sanya city and the west part of Lingshui. To the north and south of hotspot I, there is a relatively high-risk area corridor (salmon color). Hotspot III occurred in the north of Hainan Province, which is located around Haikou and Qiongshan, and also covered five counties.

## Patient distribution and risk analysis

Among the 94 IFA-positive patients, 38 were from Qiongzhong and 25 from Haikou City, and the patients from these two cities comprised 67.02% of our study population (Fig 2A). Among all the 52 WB-positive patients, 28 were confirmed from Qiongzhong People's Hospital and 12 from the Third People's Hospital of Haikou (Fig 2B), which had positive rates of 11.97% and 2.52%, respectively.

The prediction of LD (Fig 3C) can be matched with the overall distribution of IFA-positive cases. Qiongzhong with the highest IFA-positive rate; this region is located in hotspot I. Haikou and Qiongshan, which have lower IFA-positive rates, were in hotspot III. 80% cases in Danzhou (8/10) were discovered in Hotspot I, even though Danzhou was the second lowest-

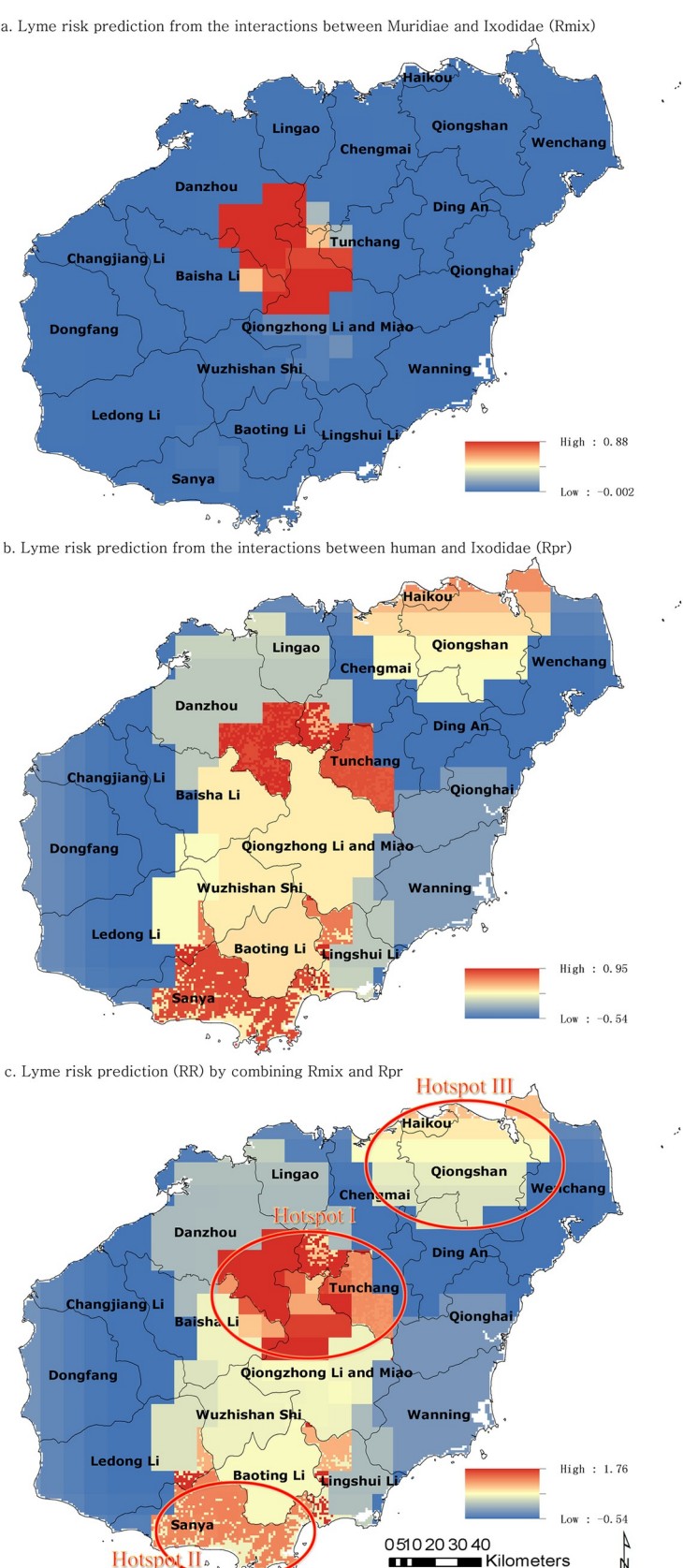

a. Lyme risk prediction from the interactions between Muridae and Ixodidae (Rmix)

b. Lyme risk prediction from the interactions between human and Ixodidae (Rpr)

c. Lyme risk prediction (RR) by combining Rmix and Rpr

**Fig 3. Prediction maps (*Rmix, Rpr,* and *RR*) of LD risk in Hainan Province.** This map was plotted by a combination of ArcGIS software version 10.1 (Esri, USA) and Photoshop CS 2.0 software (Adobe Systems, USA). The land-use raster was generated from land cover layer (http://www.resdc.cn/data.aspx?DATAID=335).

positive rate region. The positive rate of Dongfang (which is not in the hotspots of Lyme disease) was lower than that of Haikou and Qiongshan. Finally, Wenchang had the lowest positive rate, and was not in the LD risk hotspots.

The most confirmed patients (42/52, 80.77%) in all the five counties in our analysis were distributed in or around these two hotspots. Among these 42 patients, 30 WB-positive patients were located in/around hotspot I and 12 patients were in hotspot III. Another 10 cases that were WB-positive were distributed in Wenchang (4/10), Dongfang (2/10), Qiongzhong (2/10), and Danzhou (2/10)(Fig 3C).

## Discussion

LD has been studied in China for at least 30 years since the 1980s [48–49]. LD research is being carried out in Hainan Province since 2005, and was neglected for hospitals in Hainan for they still cannot do the lyme test. Clinical test of Lyme disease could tell how many Lyme patients were there in Hainan and what were their distributions. In our investigation, there were 94 positive LD cases by IFA, suggesting that 5.94% of patients with arthritis and neurological diseases were suspected to have LD. A total of 52 positive cases were identified by WB, confirming that LD is present in central and north Hainan Province. There were Lyme diseases in all 5 areas in our investigation. Lyme disease mostly distributed in temperate zone, was tested prevalent in tropical area in China. Clinicians should pay more attention of Lyme disease in this area.

According to this survey, the results showed that the infection rate (5.94%, 94 of 1583 serum samples tested positive by IFA) of LD was close to the average positive rate (5.06% [23]) of *B. burgdorferi* in the Chinese mainland. The age interval was 13–86, and there were 54 men and 40 women. The results of the WB assay verified 52 positive cases, including 30 men and 22 women (17–85 years). The results showed that LD was present in patients with arthritis or neurological symptoms in Hainan, and this disease should be commonly considered by clinicians with patients presenting with these symptoms.

In our study, the positive rate of LD was high in the central regions of Hainan and relatively low in the northern region of Hainan (Fig 2A). Of all the five hospitals included in this research, Qiongzhong People's Hospital, located in Qiongzhong City in the center of Hainan Province, had the highest positive rates of LD. In our research, we assumed that the LD infection rate of these five hospitals could represent local conditions as all these five hospitals were the typical ones in their respective local areas. Therefore, Qiongzhong had the highest prevalence of LD in Hainan Province in our research. Haikou city, which is located in the north of Hainan Province, came second in terms of the *B. burgdorferi* positive rate, followed by Danzhou and Dongfang, and Wenchang, which had the lowest prevalence of LD.

The distribution of Muridae, which is the most important reservoir of *B. burgdorferi*, could lead to a potential LD epidemic in the region[50]. Simultaneously, the interactions between Muridae and Ixodidae could help the antigen persist in its natural environment. The prediction map (*Rmix*) was overlapped by the hotspot distribution of Muridae and Ixodidae. The overlapping areas of distribution of these two genera facilitate their interactions, leading to the persistence and prevalence of *B. burgdorferi*.

Although host and vector can influence the distribution of the epidemic area[51], variation in human behavior across different landscapes may partly explain the human infection pattern

[52]. An important detail in this study is determining the most probable location where people became infected. Previous research suggests possible relationships between LD risk and human incidence, and identified areas of substantial uncertainty in relationships between tick density and human exposure to infection. Therefore, in risk analysis of the interaction between human and Ixodidae, we classified human behavior into three activities: farming, husbandry, and forestry. Rural people who engage in these activities might encounter ticks in the farmland, grass and bush, and forest areas. We calculated the sum of these risky behaviors to map the interaction between human and Ixodidae. *Rpr* reflected the risk from human contact with ticks. In Hainan, the midline region from the central to the south and the Haikou-Qiongshan area were found to be associated with the highest risk of LD infection.

The overall Lyme risk map (*RR*) that result from our analyses revealed high-risk areas for LD in the Wuzhi Mountain (hotspot I), Sanya (hotspot II) and Qiongshan (hotspot III) areas, which results from the convergence of several known risk factors, including distribution hotspot of Muridae and Ixodidae; dense rural populations; area of farmland, grassland and bush, and forest; and a dense population of cows and goats.

The three hotspots that emerged from our analysis differed in their size and overall severity. Hotspot I contained parts of five counties, which were north of Qiongzhong Li and Miao, south of Danzhou, southwest of Chengmai, west of Tunchang, and east of Baisha. Hotspot I was associated with the Wuzhi and Limu mountains, which are characterized by high densities of human, Ixodidea, and Muridae populations. In addition, hotspot I have extensive bush, grass, and forest land, which facilitate interactions between vector and potential hosts, as well as human and vector. Although hotspot II and III lack a population belonging to Muridae and Ixodidae, and feature smaller bush, grass, and forest than hotspot I, they emerged as high priority area due to their high rural population density, intensive farming, husbandry, and forestry activities. Furthermore, hotspot II should be paid more attention to as there was lack of investigation of human Lyme disease. Lyme disease might be a severe problem locally.

The current spatial pattern of Lyme risk factors is due to the interaction of humans and natural systems related to the host, vectors, and humans[53]. Therefore, we reason that it is possible to exert some control over the transmission and prevalence of LD by regulating the interaction between the relevant human and natural systems. Instead of classical statistical methods, a geographically explicit, knowledge-based method was applied here. This method was motivated by the lack of comprehensive LD data and inexplicit of complicated mechanisms of interaction of host-vector-human. Furthermore, this method not only serves as a substitute for the traditional investigation of LD, but it is also an additional tool to map risks, and it may help efforts to contain disease outbreaks and transmission. As our understanding of LD ecology improves, we can introduce new risk factors to improve our predictions of risks in space.

## Supporting information

**S1 Fig. Model Information of *Rmix* calculations.**
(TIF)

**S2 Fig. Model Information of *Rpr* calculations.**
(TIF)

**S3 Fig. Model Information of *RR* calculations.**
(TIF)

**S1 Table. The attribute of Landcover map and reclassification remap table of Landcover layer.**
(DOC)

**S2 Table. Reclassification remap table of Land-use layer.**
(DOC)

**S3 Table. The Distance Band results table using Multi-Distance Spatial Cluster Analysis tool (Ripleys K).**
(DOC)

**S1 Data. Information of Muridae distribution in all 19 cities.**
(XLS)

**S2 Data. Information of Ixodidae distribution in all 19 cities.**
(XLS)

**S3 Data. Results of Muridae Hotspot analysis.**
(XLS)

**S4 Data. Results of Ixodidae Hotspot analysis.**
(XLS)

## Acknowledgments

We are very grateful to Dr. Guo Zhongwei and Dr. Liu Xuan (Institute of Zoology, Chinese Academy of Sciences) for offering us valuable help in spatial analysis.

## Author Contributions

**Conceptualization:** Lin Zhang, Qin Hao.

**Data curation:** Lin Zhang.

**Formal analysis:** Lin Zhang.

**Funding acquisition:** Xiong Zhu, Qin Hao.

**Investigation:** Lin Zhang, Xiong Zhu, Xuexia Hou, Huan Li, Xiaona Yang, Ting Chen.

**Methodology:** Lin Zhang, Xiong Zhu, Xuexia Hou, Xiaona Yang, Xiaoying Fu, Guangqing Miao, Qin Hao, Sha Li.

**Resources:** Xiong Zhu, Huan Li, Ting Chen, Xiaoying Fu, Sha Li.

**Software:** Lin Zhang.

**Supervision:** Qin Hao.

**Validation:** Lin Zhang, Guangqing Miao.

**Visualization:** Lin Zhang, Xiaona Yang.

**Writing – original draft:** Lin Zhang.

**Writing – review & editing:** Lin Zhang, Qin Hao.

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
