## [Decision Letter · Decision Letter 0]

26 Feb 2020

Dear Dr Zhang,

Thank you very much for submitting your manuscript "Prevalence of Lyme Disease in Hainan Province" for consideration at PLOS Neglected Tropical Diseases. As with all papers reviewed by the journal, your manuscript was reviewed by members of the editorial board and by several independent reviewers. In light of the reviews (below this email), we would like to invite the resubmission of a significantly-revised version that takes into account the reviewers' comments. 

Please carefully address all the reviewer comments.

We cannot make any decision about publication until we have seen the revised manuscript and your response to the reviewers' comments. Your revised manuscript is also likely to be sent to reviewers for further evaluation.

Sincerely,

Peter J. Krause

Deputy Editor

Andrew Azman

Deputy Editor

Please carefully address all the reviewer comments.

Reviewer's Responses to Questions

**Key Review Criteria Required for Acceptance?**

**Methods**

-Are the objectives of the study clearly articulated with a clear testable hypothesis stated?

-Is the study design appropriate to address the stated objectives?

-Is the population clearly described and appropriate for the hypothesis being tested?

-Is the sample size sufficient to ensure adequate power to address the hypothesis being tested?

-Were correct statistical analysis used to support conclusions?

-Are there concerns about ethical or regulatory requirements being met?

Reviewer #1: See summary and general comments below

Reviewer #2: The authors need explain carefully on how you linked hospital data to your environmental predictors, which is very important to evaluate the performance of your predictive models. The present methods only focused much on the process of testing the disease in the lab. What is even more important is to provide the steps linking these test data with environmental climatic niches and human activities, which are unfortunately lacking here. 

Risk factors

There is no any fundamental information on how you selected the 17 cities as your study sample areas. The explicated geographical positions and occurrence of the Lyme disease reporting should also be provided here. 

The authors need explain the reason why you chose the three factors here. In addition to human activities and host characteristics, fundamental niches such as climate and microhabitat factors determine the suitability of the disease survival (e.g., Guernier et al. 2004, Liu et al. 2013). Just as what I have suggested on the Introduction section, the authors need strengthen the hypotheses around disease prevalence.

Ref.

Guernier V, Hochberg ME, Guegan JFO. (2004) Ecology drives the worldwide distribution of human diseases. PLoS Biol. 2, 740–746.

Liu X, Rohr JR, Li Y (2013) Climate, vegetation, introduced hosts and trade shape a global wildlife pandemic. Proceedings of the Royal Society B: Biological Sciences, 280: 20122506.

“were no data…” should be revised to “included no data…”

[27] red color?

One major issue in the Data analysis and Result sections is that the authors did not provide any statistical analyses to evaluate the performance of your predictive model. Although the GIS spatial analysis is fine with me, there are at least some approaches such as randomization test to validate the robustness of your predictor selection and overall predictive power of your model.

Fig. 3 please add the place name in the map to facilitate the understanding of spatial distributions of Lyme hotspots.

Reviewer #3: The study makes claims beyond its testable hypothesis. The design is inappropriate to the stated objectives -- or rather, a risk index could be valuable but this one is slapdash. Limitations on the population and sampling distribution are not addressed. No statistical analysis supports the conclusions drawn (see Summary & General Comments response).

**Results**

-Does the analysis presented match the analysis plan?

-Are the results clearly and completely presented?

-Are the figures (Tables, Images) of sufficient quality for clarity?

Reviewer #1: See summary and general comments below

Reviewer #2: I suggest shorting the paragraph on Lyme test, which is not related with your main model analysis. By contrast, the authors should pay more attentions to clarify the relationship between your hospital test and modeling analyses.

Weather should be whether

The paragraph of “as we know” is weak in logic and utility, which should be modified using more clear hypotheses driving the outbreak and prevalence of the Lyme disease. The present writing did not provide readers enough information on the background of factors important to Lyme prevalence in your study area, China and other regions worldwide. 

It is better to introduce a series of abiotic, biotic and human activities in determining the disease prevalence. This information is very helpful to represent the three predictors used in the following GIS models. In addition, the authors need strengthen the distributional patterns and crucial predictors of disease in a general way (i.e., not specific to the Lyme but other related neglected tropical bacterial diseases worldwide before talking about the Lyme disease.

As the disease widely distributes across the mainland of China, the following paragraph should stress more on the important implication of conducting Lyme disease risk analyses in Hainan province of China. I am not able to follow the present description only addressing the unique geographical position and climate. The authors need explain more on the unique opportunity of conducting this research in Hainan province, China. Only the lack of related study in Hainan is not a reasonable ground of a novel study. Furthermore, the authors need to provide readers some background knowledge on the differences of disease prevalence between mainland and island, which will make the present work more interesting and general.

Reviewer #3: The results and figures are not presented clearly (see Summary & General Comments response).

**Conclusions**

-Are the conclusions supported by the data presented?

-Are the limitations of analysis clearly described?

-Do the authors discuss how these data can be helpful to advance our understanding of the topic under study?

-Is public health relevance addressed?

Reviewer #1: I'd suggest being explicit about the limitations.

Reviewer #2: The present conclusion is too descriptive and should be strengthened by more important implications based on their main findings.

Reviewer #3: More ambitious conclusions are claimed than can be supported by the data and analysis (see Summary & General Comments response).

**Editorial and Data Presentation Modifications?**

Reviewer #1: See summary and general comments below

Reviewer #2: I have no specific comments on this field.

Reviewer #3: Significant grammar and spelling errors throughout.

**Summary and General Comments**

Reviewer #1: The investigators have undertaken a Lyme seroprevalence study in patients with neurological and/or arthritic symptoms in Hainan Province China. The findings —as reported— suggest modest prevalence 

Language, phrasing, grammar, spelling etc. 

I strongly suggest review editorial review. Unfortunately, this detracts from the manuscript considerably. Parts of the manuscript are unclear. Areas that warrant attention include —but are not limited to the following—

• Abstract: significant language issues

• “Lyme disease is one of the most important neglected tropical bacterial diseases” is it a tropical disease?

• nuerological symptoms: change “neurological”

• “single serologic test is partially hampered by the occurrence stage of antibodies”

• detecting, leading to insufficient results,

• “Weather there is risk”

• “eaqual”

• “avtivity”

• “As we all know,” would remove non-contributory statements

Methods

• How were the hospitals selected? Include brief description of the size and scope of those hospitals. Similarly, it is important to understand the representativeness of the sampling approach. How was the sample size determined?

• Need more detail about the selection/inclusion criteria i.e. “arthritis and nervous system disease”? How long had they been symptomatic? Any further clinical detail e.g. history of rash? Duration of symptoms also impacts interpretation of the results

Discussion

• Provide more information on local tick vectors and reservoir hosts

• Add a limitations section

• Could provide a few sentences on the clinical/public health implications of Lyme disease, if left neglected

• Try to contextualize the prevalence findings by offering a few sentences on the regional and/or international estimates

• Is there evidence of other tick-borne illnesses in Hainan? Have other agents been investigated locally and/or in China e.g. Babesia, anaplasma, Borrelia miyamotoi?

• Resolution of Fig 1 map needs to be improved

Tables and Figures

• Figures need legends

• Supplementary table needs a legend and explanation

Reviewer #2: Zhang et al. used 1,583 serum samples across 5 hospitals of Hainan province, China to explore the prevalence and potential predictors of Lyme disease. The topic fits well within the scope of PNTD, and the sampling effort at a province scale is impressive. In addition, Hainan province is very unique in its biodiversity and biogeography in China, and thus understanding the prevalence and the risks of important neglected tropical bacterial diseases is very timely and crucial for the prevention and control of the disease. However, my major concerns are that the current manuscript is too descriptive and should be strengthened especially for the hypothesis and additional data analyses to validate the performance of your predictive models. In addition, the scientific writing could be improved with an aid of a native speaker.

Reviewer #3: Software and risk factors

paragraph 1: More needs to be said about “all data were entered into ArcGIS as raster layers.” One almost never creates raster layers from scratch. Were they derived from satellite imagery? Converted from vector data? What does “entered” mean here?

paragraph 2: The authors likely mean “land cover” instead of “land use.” The source of the land cover data should be provided, as should a source or method of generating the “tick species distribution” layer. A description of what went into the “Tick Active Area” and “Tick Risk Area” layers is needed, as is some rationale supporting the reclassification into 9 classes. Why, for example, do the authors consider farmland a greater risk than bush? Why does seaside wetland appear to be rated higher risk than forest? It is probably not necessary to list the number of pixels containing each land cover type.

paragraph 3: “As tick questing ask for a certain height.” is a typo? Last sentence is virtually a duplicate of the last sentence of paragraph 2 and is unnecessary.

Paragraph 4: The reclassification scheme could be better conveyed using a remap table. What made the authors decide that “more tick species could increase the risk of lyme disease” – as opposed to greater tick population or greater chance of human exposure? Should be supported with citation to past research if this is true. 

Analyses

This section needs considerable expansion, perhaps to include any meaningful spatial analysis at all. How was the Tabulate Area output used and interpreted? What were the grounds for deciding that those were high risk areas? What was the means of inference to other areas? Are “cities” the polygonal regions shown in the map? 

Figure 1

This is certainly not a map of 1583 individuals. There are about 200 points in the map, and they are not well-distributed throughout the island. Perhaps the large point symbols obscure multiple individuals, in which case a smaller symbol should be used. Also some reference to the underlying population distribution would be helpful. Why are the areas with nearly no points so sparse – does no one live there? Also a description of the process used to geocode addresses is needed.

Figure 2 – A

Because of sampling bias, this map could easily be interpreted merely as a map of which districts have a testing hospital in them. The legend is badly done. It is misleading to show 6 decimal places on numbers with so little precision. The legend should be more explicit as to the quantity represented by the color ramp (it’s the IFA positive rate). Why map this instead of the 2-step positives?

Patient Distribution and Risk Analysis

Paragraph 2: “According to the survey, there were 12 species ticks in Qiongzhong (Supplemental material, S1 Table ), which is the area with most tick species in Hainan.” According to the S1 Table, there were also 12 tick species in Ledong and 10 in Sanya, both of which have no positive cases (and nearly no samples). How do the authors account for these counter examples to their argument (probably sampling bias)? Would they expect high values here if they could now acquire samples? Perhaps that would be way to test their theory.

 “In all 5 areas in our research, the spread of Lyme diseases were basically matched with the tick distribution.” The word “spread” implies an expansion over time, which is unsupported by the rest of the work. A map of the tick distribution (the number of species present in each district?) might help. 

Figure 3

There does not seem to be much predictive about Figure 3. We see the same 9 classes described above . . . are we just seeing land cover with numeric labels? If not, what went into this index? Is it an average of the rather arbitrary scores assigned to land cover, the reclassified number of tick species per district and . . . . what? Still don’t know at this point what “Tick Active Area” and “Tick Risk” are. 

Nothing in the paper supports the drawing of those arrows, as if the authors intend to show past or future change in the spatial pattern of the disease.

Discussion

paragraph 6: Similarly, there is nothing to support the verb “radiated” implying spread over time “The risk of lyme disease in Hainan was radiated from Wuzhi Mountain in the east, south and west directions . . . ”

“In the northwest of Wuzhi Mountain, there is Limu Mountain which hampered the distribution of lyme disease. “ If mountain barriers are this effective, perhaps they should be included in the risk index.

PLOS authors have the option to publish the peer review history of their article (what does this mean?). If published, this will include your full peer review and any attached files.

Reviewer #1: No

Reviewer #2: No

Reviewer #3: No
---

## [Decision Letter · Decision Letter 1]

8 Sep 2020

Dear Dr. Hao,

Thank you very much for submitting your manuscript "Prevalence and prediction of Lyme Disease in Hainan Province" for consideration at PLOS Neglected Tropical Diseases. As with all papers reviewed by the journal, your manuscript was reviewed by members of the editorial board and by several independent reviewers. In light of the reviews (below this email), we would like to invite the resubmission of a significantly-revised version that takes into account the reviewers' comments. 

We cannot make any decision about publication until we have seen the revised manuscript and your response to the reviewers' comments. Your revised manuscript is also likely to be sent to reviewers for further evaluation.

Sincerely,

Peter J. Krause

Deputy Editor

Andrew Azman

Deputy Editor

Reviewer's Responses to Questions

**Key Review Criteria Required for Acceptance?**

**Methods**

-Are the objectives of the study clearly articulated with a clear testable hypothesis stated?

-Is the study design appropriate to address the stated objectives?

-Is the population clearly described and appropriate for the hypothesis being tested?

-Is the sample size sufficient to ensure adequate power to address the hypothesis being tested?

-Were correct statistical analysis used to support conclusions?

-Are there concerns about ethical or regulatory requirements being met?

Reviewer #2: The revised method section is much more improved. Especially it has been much clearly for the part of spatial analyses. My final suggestion is that the authors need explain a bit more on the screening process of the host distribution data from GBIF. GBIF database includes occurrences in suspicious locations (e.g., on the grounds of a museum, the epicenter of a city, etc.), which are needed to remove. These records are usually incomplete sightings/specimens that are re-labelled later by people who don’t know where the sighting/collection took place.

Reviewer #3: The revised analysis is more sound than the original, but some methodological ambiguities and weaknesses remain. The objectives are articulated and the study design is appropriate, but details of the execution may lead to inaccurate results.

-- One flaw remaining in the spatial analysis portion of the revised paper is the source of the land use data. The authors cite Wikipedia and claim it was downloaded from there. I am unable to find any land use raster at the link provided, and Wikipedia is generally not in the business of doing land use classification of satellite imagery. If the authors found a dataset there, it must have come with a reference to a primary source. This matters for two reasons: for confidence in the risk analysis, it can be helpful to know how the land uses were derived and what categories were contemplated, and it is essential to know the cell size of the original to evaluate how much resampling was needed in the 10km grid analysis (for example in the computation of Cfa, Ch, and Cfo). Land use is usually derived from land cover, and categories like “sparse woods, seaside wet lands, slope grassland” sound more like land cover. The authors might consider a description of the land use / land cover data (but applicable to whatever they in fact used) like “The land use data were derived from the 2020 MODIS Terra+Aqua Combined Land Cover product, Land Cover Type Yearly L3 Global 500m, June 2020, which classifies land cover according to the International Geosphere-Biosphere Programme 17-class scheme. Land cover types were combined and reclassified into [some number of] land use categories following [some citation to the work used to inform their land use classification].” 

— The separate components of the LUmix layer are clear enough, but include the method of their final combination into one layer. A map of the LUmix layer would be good to include, as it is an important component of the spatial analysis.

— The authors make a somewhat light argument in passing for the use of a 10km grid for the rodents (searching for food), but they don’t defend its use for ticks. Is it a reasonable grid size on which to see tick clustering and habitat extent? Some supporting citation to ecological literature on gridding species ranges would be valuable here.

— The hotspot analysis seems fairly sound, but the authors might explain what hot spots/Getis G contributes that their original spatial join count of observations to grid cells does not. Why is the degree of spatial clustering more informative than just high incidence in this case? Similarly, they should explain why Fuzzy Membership is used instead of, say, simply converting to z-scores for commensurability. 

-- It would be more informative to the user to identify maximum DiffK value produced by the Ripley’s K analysis, maybe even reproduce the output table. Some explication of the meaning or interpretation of the Getis-Ord statistic, beyond the basic formula, would be useful. Also cite the ESRI documentation from which this section derives, and perhaps include the output so the reader can determine if statistically significant clusters were found at this resolution. The authors appear to be using a particular (but valid) special case of the Getis-Ord Hotspot Analysis, in which the spatial clustering is solely location-based, not weighted by the distribution of any particular attribute. Therefore, "where xj is the attribute value of Muridae" is probably not accurate. “[H]ow you construct the Analysis Field determines the types of questions you can ask. Are you most interested in determining where you have lots of incidents, or where high/low values for a particular attribute cluster spatially? If so, run Hot Spot Analysis on the raw values or raw incident counts. . . . Alternatively (or in addition), you may be interested in locating areas with unexpectedly high values in relation to some other variable. “ http://resources.esri.com/help/9.3/ArcGISEngine/java/Gp_ToolRef/Spatial_Statistics_tools/how_hot_spot_analysis_colon_getis_ord_gi_star_spatial_statistics_works.htm

— It is essential to run the Ripley’s K and Hotspot Analysis in a local projected coordinate system. The authors should disclose the projection in which they conducted their analysis.

— Rmixi =Gm*Gix*LUmix, The authors should consider and explain the effect of negative values and zeroes in this multiplication formula. What happens when Gm is close to -1 (rodent dispersion, not clustering) and Gix is close to -1 (tick dispersion)? A misleading high positive value is produced. (On review of the supporting material, it looks like the authors used Fuzzy Overlay to make this combination, but (1) why? and (2) which? Fuzzy Product sounds problematic in the ESRI documentation and (3) how would Fuzzy Overlay handle the concern about negative values?)

— The authors should consult and cite some resources on designing risk analysis formulae. 

— A source is needed for the human population density raster. Also consider that the most densely populated areas (urban) are probably at the least risk.

— “Dpri is risk from small mamal-tick interactions within the ith cell” This seems like a typo. Dpr was population density, and Rpr was a risk index that didn’t involve the mammals (rodent).

-- It would be interesting, but probably not essential, to include some sort of model validation, formalize how or whether the predicted risk aligned with the discovered cases.

**Results**

-Does the analysis presented match the analysis plan?

-Are the results clearly and completely presented?

-Are the figures (Tables, Images) of sufficient quality for clarity?

Reviewer #2: The revised result section is clear and not overstated.

Reviewer #3: The figures are NOT of sufficient quality for clarity. Values in map legends and labels on maps are too blurry to read.

**Conclusions**

-Are the conclusions supported by the data presented?

-Are the limitations of analysis clearly described?

-Do the authors discuss how these data can be helpful to advance our understanding of the topic under study?

-Is public health relevance addressed?

Reviewer #2: The conclusions are supported by the results.

Reviewer #3: The conclusions are improved in scope. They appropriately describe the risk of encounter and its relationship to the observed incidence of disease. If the methodology can be strengthened to give better confidence in the derived results, this should be an informative paper.

**Editorial and Data Presentation Modifications?**

Reviewer #2: I think the manuscript is ready for acceptance after minor revision.

Reviewer #3: The paper is still rife with English grammar errors and needs to be proofread.

"In rural area, people get to contact with ticks most likely happened when farming, husbandry or forestry activities go through."

It's also possible that some of the more grammatical and stylistically unique sentences are taken verbatim from other works and should be encased in quotation marks, e.g.

"For persistence of lyme borreliosis in the field, a seasonal synchronicity among the questing stages of vectors and the abundance of reservoir hosts is also necessary."

**Summary and General Comments**

Reviewer #2: I have no more comments here.

Reviewer #3: I don't think new experiments are needed per se, just some serious strengthening of the execution and documentation of the existing ones. Perhaps the authors might reconsider their multiplicative risk indices based on further review of the literature.

PLOS authors have the option to publish the peer review history of their article (what does this mean?). If published, this will include your full peer review and any attached files.

Reviewer #2: No

Reviewer #3: No
---

## [Decision Letter · Decision Letter 2]

18 Jan 2021

Dear Dr. Hao,

We are pleased to inform you that your manuscript 'Prevalence and prediction of Lyme Disease in Hainan Province' has been provisionally accepted for publication in PLOS Neglected Tropical Diseases.

Best regards,

Peter J. Krause

Deputy Editor

Andrew Azman

Deputy Editor

Change "the tropical area" to "tropical areas" in the Background of the Abstract.

Consider reviewer #3 comments.

Consider including one map of China to show Hainan Province in relation to mainland China

Reviewer's Responses to Questions

**Key Review Criteria Required for Acceptance?**

**Methods**

-Are the objectives of the study clearly articulated with a clear testable hypothesis stated?

-Is the study design appropriate to address the stated objectives?

-Is the population clearly described and appropriate for the hypothesis being tested?

-Is the sample size sufficient to ensure adequate power to address the hypothesis being tested?

-Were correct statistical analysis used to support conclusions?

-Are there concerns about ethical or regulatory requirements being met?

Reviewer #2: The revised methods are clear and robust.

Reviewer #3: The methods of the study are appropriate and have been extensively clarified. One remaining minor criticism is the choice of coordinate system: GCS_WGS_1984 is a geographic coordinate system, not a projected one designed to preserve distance or area. The small extent of the study area makes it unlikely that distance errors would be too substantial, but some geographic dilettantes might dismiss the hotspot analysis entirely based on this detail. I don't think it invalidates the work, but the authors might want to add a comment that indicates they are aware of the issue and made a conscious choice (and why).

**Results**

-Does the analysis presented match the analysis plan?

-Are the results clearly and completely presented?

-Are the figures (Tables, Images) of sufficient quality for clarity?

Reviewer #2: The revised results are clear.

Reviewer #3: The results are consistent with the analysis. In my pdf reader, the map text was still illegible despite the authors' efforts to increase the resolution.

**Conclusions**

-Are the conclusions supported by the data presented?

-Are the limitations of analysis clearly described?

-Do the authors discuss how these data can be helpful to advance our understanding of the topic under study?

-Is public health relevance addressed?

Reviewer #2: The revised conclusions are supported by the data.

Reviewer #3: The authors have done a good job in keeping their conclusions modest and consistent with the results. They have made explicit the sources of uncertainty in their analysis.

**Editorial and Data Presentation Modifications?**

Reviewer #2: I suggest acceptance.

Reviewer #3: The authors might change the NoData color in Fig. 2 Map A, because it is difficult to discern the difference in the two darkest greens. Perhaps NoData could be symbolized in an unrelated hue like gray. There are still some errors in written English, but they are not bad enough to impair communication.

**Summary and General Comments**

Reviewer #2: The authors have conducted intensive revisions on the manuscript. After careful reading, I find the revised analysis is clear and robust. The scientific writing has also been greatly improved. The previous reviewers' comments have been well addressed and I do not have further comments and questions. I think this is an important and timely topic, and is suitable for publication in PLOS NTDs.

Reviewer #3: (No Response)

PLOS authors have the option to publish the peer review history of their article (what does this mean?). If published, this will include your full peer review and any attached files.

Reviewer #2: No

Reviewer #3: No

---

## [Editor Report · Acceptance letter]

14 Mar 2021

Dear Dr. Hao,

We are delighted to inform you that your manuscript, "Prevalence and prediction of Lyme disease in Hainan province," has been formally accepted for publication in PLOS Neglected Tropical Diseases.

Best regards,

Shaden Kamhawi

co-Editor-in-Chief

Paul Brindley

co-Editor-in-Chief
